# Exploration of the Experiences of Persons in the Traumatic Spinal Cord Injury Population in Relation to Chronic Pain Management

**DOI:** 10.3390/ijerph20010077

**Published:** 2022-12-21

**Authors:** Tammy-Lee Williams, Conran Joseph, Lena Nilsson-Wikmar, Joliana Phillips

**Affiliations:** 1Department of Physiotherapy, University of the Western Cape, Cape Town 7535, South Africa; 2Division of Physiotherapy, Stellenbosch University, Stellenbosch 7602, South Africa; 3Department of Neurobiology, Care Sciences and Society, Karolinska Institutet, 17177 Stockholm, Sweden

**Keywords:** chronic pain, management, traumatic spinal cord injury, experiences, challenges

## Abstract

Chronic pain amongst individuals with traumatic and nontraumatic spinal cord injury (SCI) has high prevalence rates, with severe impact on the activities of daily living, mood, sleep and quality of life. This study aimed to explore the experiences and challenges of chronic pain management amongst the traumatic spinal cord injury (TSCI) population in the Western Cape region of South Africa. A qualitative descriptive approach was chosen for the study, in which 13 individuals living with TSCI were purposively recruited and interviewed telephonically. An inductive thematic analytic approach was used. The results indicate ineffectiveness of standard pain management, with a lack of education regarding pain physiology and pain management strategies as well as unbalanced decision-making between clinician and patient. Thus, patients develop coping strategies to survive with pain. Current pain regimes are suboptimal at best, underpinned by the lack of clarity or a mutually agreed plan to mitigate and eradicate pain. There is a need for chronic pain management beyond pharmacological prescription. Future practices should focus on adopting a holistic, biopsychosocial approach, which includes alternative pain therapy management. In addition, advances in pain management cannot be achieved without adopting a therapeutic alliance between the clinician and patient.

## 1. Introduction

The pooled incidence of traumatic spinal cord injury (TSCI) in developing countries was recently captured as 22.55 per million, per year [1]. In the Western Cape region of South Africa specifically, the incidence of persons suffering from TSCI was documented in a prospective study by Joseph et al. [2]. In a 1-year period, 145 cases of persons with TSCI were identified, and the crude incidence was calculated at 75.6 per million [2]; an estimate that appears to be amongst the highest in the world. The prevalence rates of chronic pain amongst individuals with nontraumatic and traumatic spinal cord injury (TSCI) have been reviewed in a meta-analysis; neuropathic pain (NeuP) occurred in 58% of studies, musculoskeletal nociceptive pain occurred in 56% and visceral nociceptive pain occurred in 20% [3]. Chronic pain also has a severe impact on activities of daily living, mood, sleep and quality of life [4,5,6,7]. Chronic pain due to SCI can include neuropathic pain, nociceptive pain and visceral pain, or a combination of these [8]. Neuropathic pain is caused by dysfunction of the nervous system and neural tissue, nociceptive pain is caused by damage to non-neural tissue, and visceral pain is caused by dysfunction of organs and is usually located in the abdominal region [9,10].

According to a review by Siddall and Middleton [11], “The effective treatment of pain following spinal cord injury (SCI) is notoriously difficult.” Sixteen years later, this is still relevant. Treatment recommendations for SCI NeuP include the use of anticonvulsant or/and antidepressant drugs followed by tramadol/opioids. Trials of medications for SCI NeuP often report limited efficacy, and even drugs shown to be effective in these studies have numerous drop-outs and report many adverse events.

A few studies have explored the views of patients in relation to their experiences with chronic pain management, and a main theme highlighted is the lack of a holistic and biopsychosocial approach to chronic pain management in the SCI population [12,13,14,15]. The literature shows that the routine focus of chronic pain management in the TSCI population consists of pharmacological care and that patients are not satisfied with this care due to its inability to reduce pain significantly, or for a prolonged period [16,17]. It is also important to note that the appropriate diagnosis of the type of pain is essential for the optimal application of pharmacological care, and that the possibility of other somatic dysfunctions should be ruled out [16]. Patients are also not satisfied with the unwanted side-effects of this pharmacological care, particularly of the central nervous system, such as dizziness and ataxia. In addition, the adverse effects extend to urinary retention, constipation and other gastrointestinal dysfunctions, sexual dysfunction and in severe cases, life-threatening hypersensitivity reactions [12,13,14,15,17,18]. Despite evidence indicating that alternative therapy has a role to play in chronic pain management, it has been found that patients with SCI have not been referred for such alternative/nonpharmacological therapy [13,14,15]. Thus, there is a need for chronic pain management beyond pharmacological care in the SCI population. In addition, chronic pain present in the SCI population can be modulated through psychological and social factors, such as perceived life satisfaction, life control and social support [19], which should also be addressed.

Healthcare professionals showed a decreased knowledge about general chronic pain management [18]. Many healthcare practitioners believed that the majority of their patients exaggerated their pain [18,20,21,22]. However, there is a lack of studies investigating the knowledge and satisfaction of healthcare professionals in relation to chronic pain management in the SCI population. One study by Norrbrink et al. [15] reports that physicians are frustrated with the lack of time and personal competence in treating chronic neuropathic pain in the SCI population. In addition, physicians expressed the need for cognitive–behavioural therapy, as well as the acceptance of and commitment to therapy to manage chronic pain where pharmacological care was ineffective. It is important to note that due to the limitations of studies assessing the knowledge, perception and satisfaction of physicians in relation to chronic pain management in the SCI population, these studies leave much to be desired with regard to the preparedness of the healthcare system to appropriately attend to the problem of chronic pain in this population.

On the other hand, studies investigating patients’ views in relation to chronic pain management have found that patients required more information regarding treatment options to allow for self-management. In addition, the effectiveness of chronic pain treatments requires long-term monitoring [12,15]. Widerström-Noga et al. [19] recommend a biopsychosocial assessment and treatment plan for effective chronic pain management as opposed to the routine pharmacological focus of chronic pain management. The biopsychosocial model was developed as early as 1977 by George Engel [23]. The biopsychosocial model focuses on illness being an interaction of biological (disorder of body structures and/or organ systems), psychological and social factors [23]. Similarly, Waddell [24] has highlighted that pain cannot be understood without understanding the individual suffering from the pain and, therefore, the interrelationship between biological processes, psychological status and sociocultural context must be considered in order to fully assess and manage pain.

Very few qualitative studies exist that investigate the perspectives of persons with TSCI in relation to chronic pain management, and to the best of our knowledge, none exist in South Africa. One quantitative, four-year follow-up cohort study found the period-prevalence of pain to be 80% amongst persons with TSCI. However, the nature, frequency, and intensity of pain, as well as experiences concerning pain, have not been explored [25]. The purpose of this study was to qualitatively explore experiences and challenges in relation to the management of chronic pain amongst the TSCI population in the Western Cape region of South Africa.

## 2. Materials and Methods

A qualitative descriptive approach was chosen for the study. This approach is best suited to describe the chosen phenomena as it provides factual responses about the experiences and challenges of participants in relation to chronic pain management [26]. Purposive sampling, with a maximum variation technique (with respect to gender, level of injury, ASIA classification, type of pain and prescription of pain-specific medication), was used to recruit 13 persons from a previous study which aimed at gathering data regarding the prevalence of chronic pain and the impact thereof in the TSCI population. No a priori sample size was determined, as per the traditions of qualitative research, but the principle of theoretical saturation was applied, whereby participants were enrolled until no more new experiences and perspectives were shared. Data for the previous study were collected from participants using the International Spinal Cord Injury Basic Pain Dataset (ISCIPDS:B).. The BPDS collects data about the type and area of pain, how often pain occurred in the last seven days and how pain interferes with activities of daily living, mood and sleep [27]. In addition, the ISCIBPDS classifies below-level neuropathic pain (BL-NeuP) as “hot, burning, tingling, prickling, pins and needles, squeezing, cold, electric or shooting more than 3 dermatomes below neurological level (NLI) and may be perceived up to NLI” [27]. Furthermore, musculoskeletal nociceptive pain (NP) is classified as “dull/aching/initiated or aggravated by movement” [27].

### 2.1. The Inclusion Criteria for the Study Are Listed Below

Adult (older than 18 years)Diagnosis: traumatic spinal cord injuryInjury occurring in the last five years, i.e., April 2016–April 2021Injury occurring in the Cape Metropolitan region of the Western Cape

The data were collected from the Western Cape Rehabilitation Centre, a specialised rehabilitation centre in the Cape Metropolitan region of South Africa. In line with the protection of personal information (POPI) act, the researchers enlisted the aid of healthcare professionals who have attended to the patients, as well as peer supporters who have a relationship with these patients, in order to contact the participants. Table 1 describes the characteristics of the participants.

### 2.2. Participants

Thirteen participants participated in the qualitative study; 11 were male and 2 were female. The mean age of participants was 40.5 (standard deviation 10.6), with a four year mean of living with a TSCI (standard deviation 0.9). 100% of participants suffered from at least one type of chronic pain, 46.2% suffered from two types of chronic pain and 23.1% of participants suffered from three types of chronic pain. BL-NeuP was the most common type of chronic pain. NP was the second most common chronic pain, occurring in three participants as their first pain as well as their second pain.

### 2.3. Data Collection

Semi-structured interviews were conducted via telephone at the University of the Western Cape. The interviews lasted 20–30 min each. After interviewing 13 persons, data saturation was reached, i.e., no new experiences and challenges emerged from the interviews [28]. Theoretical saturation was underpinned by the variation of participants in terms of demographic, injury, and contextual factors, to ensure that a broad range of experiences and perspectives were sought.

Each interview was recorded. The questions included:What are the medications’ names that you are currently using? Prompts: how many times do you take it in a day? (Only when you have pain or as prescribed, for example three times a day)How else do you manage your pain? Prompts: nonpharmacological (name the type of nonpharmacological treatment and how often it is performed).Are you satisfied with your pain management? Prompts: Does the medication take your pain away or improve your pain? How long does the pain stay away for?How do you cope with your pain? Prompts: What do you do to ease your pain other than the management described above?What education did you receive regarding pain and pain management?What advice would you give to health professionals about pain management?

Prior to the interviews, a medical records study of each participant was done at the Western Cape Rehabilitation Centre in order to extract medication details as well to confirm the presence of a TSCI.

### 2.4. Data Analysis

All interviews were transcribed professionally. Interviews conducted in a different language were translated professionally following transcription procedures. The transcripts were imported to Atlas.ti, version 22 software for coding and data management. The researchers (T.W & J.P) familiarised themselves with the content by reading through the transcripts several times. An inductive thematic analysis was incorporated whereby the data informed the emerging themes. Ideas and themes from the manuscripts were captured and a coding system was created as data were marked when they related to one or more of the themes [28].

The trustworthiness—representing aspects of validity and reliability in qualitative research—of the data was ensured through various measures, such as giving a few of the participants a summary of the data captured to verify whether the data were interpreted correctly according to their opinion. A description of the setting, methodology and data analysis process are provided for the reader to decide whether the setting and findings are similar to another setting with which they are familiar, in order to apply the findings to their setting. In addition, these processes are provided for researchers to be able to repeat the study [29].

## 3. Results

Figure 1 is a representation of the main theme, the themes and subthemes. The main theme that emerged from the interviews related to the discontentment and frustration participants experience with chronic pain management. The results suggest that participants are experiencing an experiential journey of learning to live with chronic pain. In addition, current pain regimes are suboptimal at best, underpinned by the lack of clarity and mutually agreed plans to mitigate and eradicate pain. There is a need for improved therapeutic alliance between the clinician and patient, to help ensure shared decision-making on treatments, which impact on the patient. Through this journey, patients have developed coping strategies, which are mere means of survival, characterised by reduced functioning frameworks, spontaneity and a need for support.

The results will further be described under the theme and subtheme headings. Please note that English is not the first language for many of the participants interviewed in this study. In addition, one participant uses the Afrikaans language to describe her experience.

### 3.1. Standard Pain Management Strategies

#### 3.1.1. Ineffectiveness of Pharmacological Management

A main concept which emerged with regard to pharmacological management was the partial effectiveness of the medication, as it only relieved chronic pain for a short period of time. In addition, many participants expressed that pharmacological management did not relieve their pain significantly or for prolonged periods. A participant felt that the root cause of the problem was not addressed, and that the pharmacological management was simply a blanket treatment.

“It only helps for a short period of time”. (P1)

“It doesn’t take my pain away because like if I drink those medications for some minutes I start to feel the pain again”. (P5)

However, a few participants found that taking the medication as prescribed, even in the absence of pain, aided in the reduction of their occurrence of pain. “I take the tablets like every day so the pain don’t happen often”. (P13) On the contrary, one participant mentioned that despite taking the pharmacological management daily, the participant still experienced extreme pain. He described the pain as “the pain is too much”. (P6)

#### 3.1.2. Nonpharmacological Management Discovered by Patients

Nonpharmacological treatment techniques discovered by the participants included exercise/walking with calipers/movement/moving around in the area with the wheelchair, menthol ointments (such as wonder cream), African Nidorella plant and cannabis usage. From these treatments, the participants reported that the exercise/walking with calipers/movement/moving around in the area with the wheelchair and menthol ointments (such as wonder cream) were effective in relieving their pain.

“Then when I do the exercises … it helps the pain because why it [exercises] can get the area that I’m exercising, that I’m moving my body”. (P8)

One participant was prescribed calipers for his lower limbs in order to aid mobility and when describing their use, the participant mentioned “If I put those things I don’t feel pain because the blood will be circulating.” (P5)

The menthol rub and wonder cream were described as being effective in relieving the pain for a short period of time, as well as aiding sleep.

“The relieve is just for a few hours and then it is back again”. (P3)

“So the moment I go to sleep then the pain is gone”. (P2)

Nonpharmacological management recommended by healthcare providers consisted of physiotherapy. Only one participant mentioned that the doctor at the local clinic referred the participant for physiotherapy, which succeeded in relieving the frequency of pain.

“So he said go see a physiotherapist and since I see the physiotherapist then the pain doesn’t come so much.” (P1)

#### 3.1.3. A Lack of Education Regarding Pain and Pain Management Strategies

A main concern which emerged from the interviews was the lack of information regarding pain physiology and pain management from healthcare professionals as well as peer facilitators. One participant in particular was disturbed by the lack of information provided to her by her surgeon regarding a procedure which was done.

“They didn’t teach me anything”. (P3)

“He just only give me the medication”. (P6)

“Like I said after the operation I didn’t see the surgeon and he didn’t come and explain to me what they did, what they saw and what was wrong and what did they put in. I didn’t have any of that information”. (P4)

However, a few participants reported that according to the education they received, the cause of chronic pain was attributed to nerve and muscular dysfunctions, such as muscular spasms. Though, when asked if the healthcare professional explained the cause of the spasms, a participant answered “No he didn’t put it in so many words.” (P2)

“It’s the nerves. I think the nerves is getting irritated because why it’s now not moving or anything so now it’s just getting the spasms”. (P13)

Education received regarding pain management techniques was only reported by a few participants and consisted of continuing pharmacological management, exercise and distraction practices. However, it was apparent from one participant that continuing exercise was not relayed as a pain-relieving technique, as seen in the quote, “No, I was not taught how I should manage my pains, but I did receive advises that I should exercise more” (P11).

“…they said I mustn’t stop my pain tablets, I mustn’t stop the nerve tablets, you know, they said it’s very important to take them but the pain will never go away. And one thing in particular I can remember what they said to me was that I must listen music in my ears to distract me from the pain, that’s all they educate me about, that’s all that they teached me”. (P8)

Participants were asked to describe the education received from peers. Participants are visited by peers who were previously admitted to the rehabilitation centre in order to share their experiences regarding life in the community following discharge. All the participants mentioned that the peer facilitators did not educate them about pain or how to manage it. The participants mentioned that these sessions were mostly motivational talks.

“No. We were actually just talking about everyday things. But nothing about that [pain education]”. (P4)

“Nothing my dear”. (P8)

A main concern regarding pain management satisfaction was that most of the participants expressed discontentment. Participants also expressed that they felt alone in their chronic pain journey, where the pain was solely their burden to carry. On the other hand, a few participants were satisfied with their pain management and attributed their satisfaction to taking their analgesics exactly as prescribed, as well as the combination of self-management and analgesics, as seen from the following quote when asked about the efficacy of the wonder cream usage: “Otherwise I think it work with the pain tablets” (P1).

“I’m not satisfied because even if I have those tablets, I keep on feeling the pain”. (P5)

“I take the tablets like every day so the pain don’t happen often”. (P13)

### 3.2. Unbalanced Decision-Making

#### Clinician–Patient Relationship

A main concern which emerged was the lack of a therapeutic alliance between patients and their clinicians. In addition, participants reported feeling insignificant when interacting with their healthcare professionals. Participants expressed the need to be heard as well as the need for their providers to show empathy. After all, the patients are the experts on their pain experiences and should offer insights into the behaviour of pain.

“When I told him about this pains in the leg and that, I told him I am the one feeling the pains not you, so he can’t tell me that he cannot make my…how can I say, he can’t make the dosage higher, he must listen to me because it’s me sitting in this wheelchair, it’s not you, so they must listen to their patients man, so that the patients can explain to them thoroughly what is going on so that they can understand what they have to do to help the patient”. (P13)

“That is why I stopped asking the doctor to actually look at these things because for it feels like they don’t really take note of it”. (P4)

Participants also felt that healthcare professionals should improve their practices, as the current management of chronic pain is not effective.

“I can say try to give more physio to the patient and see what is the outcome of it. That is the best option I can see”. (P2)

“You know what, to be honest with you, I think the doctors can get, they can do better to help us with our pain, really you know, really they can do better”. (P8)

### 3.3. Coping Strategies

Participants were asked how they cope with their pain, as well as which factors they rely on in order to persevere despite their pain and circumstances. The subthemes identified included behavioural adaptations and cognitive adaptations as well as internal and external sources. These coping strategies are implemented as a means of survival.

#### 3.3.1. Behavioural and Cognitive Adaptations

A few participants described behavioural adaptions, which consisted of participants moving away/changing their aggravating positions to relieve their pain. In addition, the main concepts which emerged regarding cognitive strategies consisted of acceptance and distraction techniques. The acceptance of one’s situation, as well as the acceptance of their pain as a chronic condition, were reported. Various distraction techniques were described and a few involved reading, playing with a pet, engaging in conversations with family and friends, watching a movie or listening to music.

“You see when I lay on my back and I put my legs up that is when the pain starts. So I don’t lay on my back anymore, I just lay on my sides”. (P1)

“I have accepted that I will live with these pains for the rest of my life”. (P11)

“I try to focus on something else. I try to read; I would try to play with Jojo [cat] or what I do is, I try to sing”. (P4)

#### 3.3.2. Internal Resources

The two main internal resources which were described by many participants to aid them in coping consisted of faith and a positive mindset. In addition, a few participants mentioned that they rely on themselves to get them through each day.

“My dear I’m praying a lot my dear. What keeps me going is to live a righteous life and to be strong in my faith you know, my God is really strengthening me and he’s giving me the peace … I’m always happy, irrespective my pain you know, so if you ask me what is keeping me going, it’s my faith in God my dear”. (P8)

“I think I lean on myself and my own strength because I know there is a lot that I can handle”. (P4)

#### 3.3.3. External Resources

Family support was the main external resource described by many participants as a coping mechanism. Many participants expressed extreme gratitude for their families, because the adaptation to a new life in a wheelchair as well as the addition of a chronic pain condition was, and still is, a challenging journey for them.

“I am just grateful to my wife and children because they are the only one who knows exactly what I go through, even though at times I snap at them or fight with my wife because this life and pains are emotionally consuming me”. (P11)

“I have a sister and with her, she keeps me going man, like when I don’t want to stand up, she say hu uh, kom, kom, ruk jou reg (Afrikaans to English translation: “no, pull yourself together”)… like I have a very supportive family so they are always there to uplift and to motivate me, you know there is that time when you think like yoh I don’t have lus today or so man, but they don’t give me time to think of that”. (P13)

## 4. Discussion

This study explored the current management of chronic pain as well as the challenges experienced amongst the TSCI population in the Western Cape region of South Africa. The main findings consisted of the following themes: (1) standard pain management strategies; (2) unbalanced decision-making; and (3) coping strategies.

Within the “standard pain management strategies” theme, various pharmacological management techniques were discussed, and many participants found this management ineffective. This finding is consistent with other studies investigating chronic pain management amongst SCI survivors, where patients have reported that they do not want to use medications due to their inefficacy as well as unwanted side-effects [14,15,18]. In addition, our study found that participants were largely unsatisfied with the current management of their chronic pain. Similarly, Murphy and Reid [16], found that amongst persons suffering from chronic nociceptive pain in the SCI population, those using ibuprofen, narcotics and hot packs were dissatisfied with their pain management due to the lack of sufficient pain control. In addition, physicians have expressed their lack of competence in treating chronic pain, in particular neuropathic pain, in the SCI population, due to a lack of time and inefficacy of pharmacological care [15]. There is also a general lack of knowledge with regard to chronic pain management amongst healthcare professionals [18]. It is evident that there is a need to expand chronic pain management in the TSCI population beyond pharmacological recommendation. A recently published guideline [30] recommends physical therapy, exercise, physiotherapeutic techniques and psychotherapeutic techniques as second-line therapy in addition to pharmacological management in the treatment of chronic neuropathic pain in the SCI population. In addition, physical therapy, exercise, physiotherapeutic techniques, psychotherapeutic techniques, massage/heat therapy and hydrotherapy are recommended as nonpharmacological management techniques. Despite these guidelines, the current study found that only one participant reported that he was referred for physiotherapy as a nonpharmacological management technique for his chronic pain. Physical activity and physical therapies, such as heat therapy, ice therapy, massage, TENS etc., have been proven to significantly reduce chronic pain in the SCI population [31,32]. Concurrently, our study found that the types of self-discovered nonpharmacological management techniques which aided in pain relief consisted of exercise and the application of menthol ointments/creams. Finally within this theme, most of the participants expressed the lack of education provided to them regarding pain physiology and pain management techniques. The literature shows that comprehensive pain management programmes, usually consisting of an educational component (pain physiology and pain management techniques), cognitive–behavioural therapy (CBT) component and relaxation exercise, result in improvements in psychological distress in the SCI population [33,34]. Chronic pain has been shown to correlate with psychosocial factors in the TSCI/SCI population [5,19,35]. Examples of the factors shown to severely impact on chronic pain and quality of life include catastrophizing, helplessness, sleep, depression and perceived social support [5,19,35]. Thus, it is as important to address psychosocial factors as it is to address the chronic pain itself. In addition, pain intensity and pain-related disability have shown improvement with the modification of pain coping cognitions and strategies through a CBT programme, in this population [36,37,38].

Within the “unbalanced decision-making” theme, it was evident that shared decision-making between healthcare professionals and patients is lacking in the treatment of patients’ chronic pain, with participants expressing their need to be heard when interacting with healthcare professionals. This is consistent with similar studies [12,15]. Shared decision-making (SDM) is defined as an interactive process between the health professional and client when making healthcare decisions. SDM is a cornerstone concept for patient-centred care [39,40,41,42]. This concept is built on the assumption that both parties have equally important information to contribute to the decision-making. Health professionals are the experts in their field of study, from diagnoses to treatment options and side-effects, whereas the clients are the experts on their own values, treatment preferences and goals [43]. SDM has been shown to improve adherence to changed behaviour, which is often required by persons suffering from chronic illnesses [44]. Persons suffering from chronic pain in the TSCI/SCI population are required to change their behaviour, especially with CBT programmes, in order to improve coping with pain, pain intensity and pain-related disability [36,37,38]. This emphasises the importance of shared decision-making, which has not been shown to be present amongst the participants in our study. This seems to be a common trend, as other studies have also found that patients expressed the need for healthcare professionals to listen to them and to respect their knowledge [13]. In addition, the desire to work with the doctor as a team was also expressed [45]. Ultimately, within the pain-recommendations theme, it was evident that persons within the TSCI population suffering from chronic pain expressed the need for alternative therapy, in particular, physiotherapy, to manage their pain. This was seen in other studies as well [12,15]. As mentioned previously, physical exercise, which is often part of overall physiotherapy management, has been proven to improve chronic pain in the TSCI population [31,32].

In the current study, acceptance and distraction techniques were found to aid participants in coping with their current situation, as well as their chronic pain. Similar findings were reported in a comparable article by Henwood et al. [18]. As mentioned previously, pain can only be understood through the complex interrelationship between biological processes, psychological status and sociocultural context, as part of the biopsychosocial approach to chronic pain [25]. Two interesting psychological or cognitive interventions which are used for the management of chronic pain are acceptance and distraction. Acceptance is defined as “…a willingness to experience continuing pain without needing to reduce, avoid or otherwise change it” [46]. Although not widely studied in the SCI population, earlier studies show that acceptance interventions are successful in reducing pain and psychological distress as well as physical and psychological disability [46,47,48]. More recent studies support the finding that acceptance improves physical and psychological functioning in persons suffering from chronic pain in general [47,49,50]. In addition, a guideline has been published for the implementation of acceptance interventions in persons with chronic pain as a whole [51]. Distraction interventions are based on the concept that the implementation of sensory distraction leaves less space for pain processing as attentional resources are already limited in cognition [52]. A recent systematic review found that poor quality articles with a wide variation in practice sessions, dose frequency and duration offered poor comparison of distraction techniques with control groups, and as such, there was no difference in effects between the treatment and control groups in persons with various types of chronic pain [53]. However, an exciting development in the last two decades is the emergence of virtual reality (VR) as a nonpharmacological management for pain. A recent article [54], providing an overview of current literature investigating VR pain therapy in persons with various types of acute and chronic pain, found promising results. To our knowledge, there are no studies investigating VR pain therapy in the SCI population suffering from chronic pain. However, various chronic pain conditions have been studied and the results showed that patients were empowered to better manage their chronic pain through distraction, focus shifting and/or building skills to moderate the processing of pain sensations. Skill building consisted of practicing relaxation through VR intervention [55,56,57,58]. However, a common limitation for VR pain therapy in patients with chronic pain is the lack of prolonged effects following the conclusion of this therapy. To conclude, VR pain therapy seems to be a worthwhile intervention which, along with other distraction interventions, should be investigated in future chronic pain studies in the SCI population.

It must be noted that participants in this study resided in low-income areas and the effect of socioeconomic status cannot be discounted when considering access to the alternative management strategies mentioned above, in particular VR pain therapy. In South Africa, equal distribution of and access to resources are commonly limited in low-income regions, due to, among other causes, health care staff shortages, availability of services at all levels of health care, and the increasing burden of disease and poverty [59,60,61]. The problem of understaffing leads to prolonged waiting periods for persons to access services such as physiotherapy and psychology in the governmental health sector. This is an issue which should be addressed at a governmental level to ensure a strong workforce to better the health and wellbeing of all citizens.

## 5. Conclusions

Persons living with TSCI and chronic pain are not satisfied with the current management of their chronic pain. Pharmacological management is not effective and no referral for alternative therapy is made available. The discontentment of current management practices is underlined by the lack of SDM between client and clinician. As a result of suboptimal pain management, patients have developed coping strategies which include acceptance, distraction, family and self-support, and movement, as well as religious practices.

Future practices should move beyond pharmacological care for chronic pain. Advances in this field should include a multidisciplinary approach, addressing biological processes, psychological status and sociocultural context outside of the specialised rehabilitation centres. In addition, improvements in current chronic pain management should be underpinned by the SDM concept.

There is a need for governmental intervention to provide equal access to both physical and human resources to aid in improving access to alternative pain management strategies for persons with TSCI in the Western Cape Region of South Africa.

Future research should assess alternative therapies for chronic pain management. In addition, randomised controlled trials should be conducted to assess acceptance and distraction techniques as interventions for chronic pain in the TSCI population.

Lastly, pain is a universal concept; however, it is understood, experienced, and mitigated differently between people, contexts, and countries. Pain management relies on diverse and comprehensive services, which may not be equal in each country. Hence, this study provides an understanding of experiences in one setting, which could provide insights into other similar or dissimilar settings.

## Figures and Tables

**Figure 1 ijerph-20-00077-f001:**
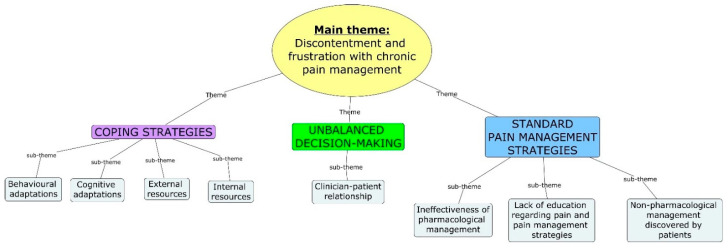
Main theme, themes and subthemes.

**Table 1 ijerph-20-00077-t001:** Description of participants, their pain and pharmacological management.

Participants (P)	Gender	Age	Year of Injury	Level of Injury and ASIA Classification	The Most Common Location and Type of Pain	The Second Most Common Location and Type of Pain	The Third Most Common Location and Type of Pain	Pharmacological Management	Nonpharmacological Management
1	Male	33	2020	T1, ASIA A (motor and sensory complete)	Below-level NeuP; bilateral feet	None	None	Baclofen (skeletal muscle relaxant) Triptyline (amitriptyline) (antidepressant and analgesic for various types of pain ranging from neuropathic to fibromyalgia to migraines and tension headaches)	Exercise
2	Male	39	2017	C5, ASIA A (motor and sensory complete)	NP; bilateral lower limbs	None	None	Baclofen (skeletal muscle relaxant) Amitriptyline (antidepressant and analgesic for various types of pain ranging from neuropathic to fibromyalgia to migraines and tension headaches) Panadol (analgesic for mild pain) Tramadol (analgesic for mild to moderate pain)	Wonder cream Moving around in the area with the wheelchair
3	Male	47	2019	T9, ASIA A (motor and sensory complete)	Visceral NP, stomach	Below-level NeuP; lower back	Below-level NeuP; bilateral lower limbs	Pregabalin/Lyrica (analgesic specifically for neuropathic pain, anticonvulsant and anxiolytic)	Menthol rub Self-massage
4	Female	54	2017	T12, ASIA C (motor and sensory incomplete)	Below-level NeuP; Genitalia	Below-level NeuP; bilateral lower limbs	Visceral NP; throat	Ibuprofen (analgesic and anti-inflammatory) Pregabalin/Lyrica (analgesic specifically for neuropathic pain, anticonvulsant and anxiolytic)	Does not use nonpharmacological management
5	Male	36	2018	T12, ASIA B (motor complete and sensory incomplete)	Below-level NeuP; bilateral lower limbs	Visceral NP; stomach	None	Baclofen (skeletal muscle relaxant) Pregabalin/Lyrica (analgesic specifically for neuropathic pain, anticonvulsant and anxiolytic) Paracetamol (analgesic for mild to moderate pain) Carbamazepine (anticonvulsant medication for epilepsy and neuropathic pain)	Walking with calipers
6	Male	61	2018	C4 (ASIA classification not documented in file)	Below-level NeuP; bilateral lower limbs	None	None	Tramadol (analgesic for mild to moderate pain) Carbamazepine (anticonvulsant medication for epilepsy and neuropathic pain)	Does not use nonpharmacological management
7	Male	30	2017	C7, ASIA A (motor and sensory complete)	NP; left shoulder	NP, bilateral lower limbs	None	The participant reports that his pain is not severe enough to use medication.	The participant reports that his pain is not severe enough to use nonpharmacological management.
8	Male	35	2018	T11, ASIA A (motor and sensory complete)	Below-level NeuP; stomach	Visceral NP; bladder	None	Pregabalin/Lyrica (analgesic specifically for neuropathic pain, anticonvulsant and anxiolytic) Triptyline (amitriptyline) (antidepressant and analgesic for various types of pain ranging from neuropathic to fibromyalgia to migraines and tension headaches) Carbamazepine (anticonvulsant medication for epilepsy and neuropathic pain) Paracetamol (analgesic for mild to moderate pain) Adco-dol (analgesic for mild pain)	Exercise
9	Male	35	2018	T12, ASIA B (motor complete, sensory incomplete)	Below-level NeuP; right lower limb	None	None	Baclofen (skeletal muscle relaxant) Pregabalin/Lyrica (analgesic specifically for neuropathic pain, anticonvulsant and anxiolytic) Valium (anxiolytic medication for anxiety, seizures, alcohol withdrawal, muscle spasms, insomnia and restless leg syndrome) Amitriptyline (antidepressant and analgesic for various types of pain ranging from neuropathic to fibromyalgia to migraines and tension headaches)	Smoking cannabis
10	Male	47	2017	T6, ASIA classification not documented in file, however, motor complete, sensory incomplete documented)	Other NP; Head	None	None	Grandpa (analgesic for mild to moderate pain) Paracetamol (analgesic for mild to moderate pain) Carbamazepine (anticonvulsant medication for epilepsy and neuropathic pain) Pregabalin/Lyrica (analgesic specifically for neuropathic pain, anticonvulsant and anxiolytic) Amitriptyline (antidepressant and analgesic for various types of pain ranging from neuropathic to fibromyalgia to migraines and tension headaches)	Does not use nonpharmacological management
11	Male	47	2018	C8, ASIA A (motor and sensory complete)	Below-level NeuP; left hip	NP; bilateral fingers	Below-level NeuP; left shoulder	Triptyline (amitriptyline) (antidepressant and analgesic for various types of pain ranging from neuropathic to fibromyalgia to migraines and tension headaches) Ibuprofen (analgesic and anti-inflammatory) Tramadol (analgesic for mild to moderate pain)	African plant called NidorellaExercise Ointment use Self-massage
12	Male	41	2018	T2, ASIA A (motor and sensory complete)	NP; upper back	None	None	Norflex (skeletal muscle relaxant) Amitriptyline (antidepressant and analgesic for various types of pain ranging from neuropathic to fibromyalgia to migraines and tension headaches) Painamol (analgesic for mild to moderate pain)	Does not use nonpharmacological management
13	Female	21	2018	T5, ASIA A (motor and sensory complete)	Below-level NeuP; genitalia	None	None	Baclofen (skeletal muscle relaxant) Pregabalin/Lyrica (analgesic specifically for neuropathic pain, anticonvulsant and anxiolytic)	Heat/Hot water bottle use

NP = Nociceptive pain; NeuP = Neuropathic pain.

## Data Availability

Not applicable.

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
