# Peer review of "Exploration of the Experiences of Persons in the Traumatic Spinal Cord Injury Population in Relation to Chronic Pain Management"

_ijerph, 2022, doi:10.3390/ijerph20010077_

Round 1
Reviewer 1 Report
The authors put forward an interesting manuscript analyzing the pain management themes in individuals with traumatic spinal cord injury. The comments from the participants and the subsequent analysis may prove useful to the research community that is unable to interact directly with the SCI community and may promote further research in this field. However, the authors sampled from a relatively small pool of participants for a public health study. A stronger justification for selecting this small number of participants is needed. Additionally, while the authors describe the importance of socio-psychological status, there is no mention of socio-economic status, which may be equally important in obtaining proper pain management treatment and information. Thus, it is my suggestion that the authors revise the manuscript to justify this lack of participants and information or to include this data in a subsequent revision of the manuscript.
Following are specific comments, questions, and suggestions:
1. Numbered sentences in the abstract are distracting and can be misconstrued as references.
2. Line 33 represents the prevalence rates of neuropathic pain as a percentage of studies, rather than individuals with SCI since the percentages are obtained from a meta-analysis of multiple studies. It is suggested that the sentence is reworded to make this clear.
3. Line 33 establishes NeuP as the abbreviation for neuropathic pain, but line 43 and 45 use SCI-NP.
4. The abbreviation for TSCI is established twice in the introduction (Lines 33 and 51).
5. Please include demographics of TSCI in the Western Cape region of South Africa.
6. Was the degree of injury considered when selecting participants via purposive sampling? A participant with a clinically incomplete C4 injury will report widely different experiences than one with a clinically complete C4 injury. In addition, this information would be valuable to note in Table 1.
7. What was the basis for selecting 13 participants specifically?
8. There are several mentions of the socio-psychological aspect of pain management. However, there is little mention of the participant’s socio-economic status. Their socio-economic state may be highly relevant in whether they are able to access some of the pain management strategies described. For example, did participant #7 not take medication due to successful pain management or due to lack of resources?
9. A general state of their mental health may also provide important information. As an example, the authors may note if participants are taking medication for depression (though some of the medications mentioned may be used as such, e.g. Valium).
10. The authors suggest using virtual reality as a possible approach to pain management. However, VR may be unattainable for members of the community with low financial resources. Please discuss more in detail the effect of socio-economic state on the SCI population’s ability to obtain effective pain management treatment.
11. Supplemental material with all the transcribed participant’s responses, rather than selected few as in the manuscript, may be useful to the scientific community.
Reviewer 2 Report
Please see attached document; thank you.

Round 2
Reviewer 1 Report
The authors have answered the reviewers' concerns appropriately and the manuscript is in adequate shape for publication.
Reviewer 2 Report
I thank the Authors for responding to my initial critique and am satisfied with their changes to the paper and accompanying rebuttal.